# Do Vision Models Develop Human-Like Progressive Difficulty Understanding?

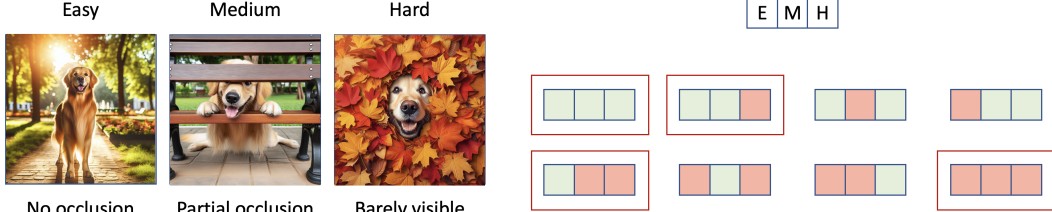

Figure 1: **Left:** Sample images from our proposed test set generated using GPT-4 + DALL-E 3. For the class of *golden retriever* and attribute *occlusion*, we generate images of varying difficulty. Intuitively, it is easier to classify the leftmost image as *golden retriever* compared to rightmost image. **Right:** Possible responses (correct/incorrect) of a model on the easy/medium/hard image on the left side. *Cannot solve easy one → cannot solve difficult one. Can solve difficult one → can solve easy one*: this hypothesis is only satisfied in 4 (in red) out of 8 possibilities.

## Abstract

When a human undertakes a test, their responses likely follow a pattern: if they answered an easy question $(2 \times 3)$ incorrectly, they would likely answer a more difficult one $(2 \times 3 \times 4)$ incorrectly; and if they answered a difficult question correctly, they would likely answer the easy one correctly. Anything else hints at memorization. Do current visual recognition models exhibit a similarly structured learning capacity? In this work, we consider the task of image classification and study if those models' responses follow that pattern. Since real images aren't labeled with difficulty, we first create a dataset of 100 categories, 10 attributes, and 3 difficulty levels using recent generative models: for each category (e.g., dog) and attribute (e.g., occlusion), we generate images of increasing difficulty (e.g., a dog without occlusion, a dog only partly visible). We find that most of the models do in fact behave similarly to the aforementioned pattern around 80-90% of the time. Using this property, we then present a new way to evaluate those models' image recognition ability. Instead of testing the model on every possible test image, we create an adaptive test akin to GRE, in which the model's performance on the current round of images determines the test images in the next round. This allows the model to skip over questions too easy/hard for itself, and helps us get its overall performance in fewer steps.

## 1 Introduction

Imagine a math teacher grading a student's answer sheet, and finds that they got the answer of $(2 \times 3)$ wrong but the answer for $(2 \times 3 \times 4)$ right. The teacher will rightly wonder whether the student properly learnt the concept of multiplication or whether they memorized the answer to the more difficult question. This is because there is a characteristic way in which humans learn any concept: if they cannot answer an easy question, they very likely *cannot* answer a more difficult one. And conversely, if they can answer a difficult question, they most certainly *can* answer an easier one as well. Neural networks are also trained to learn concepts to perform a task. Do they also learn those concepts in a similarly characteristic way?

In this work, we study this question for the task of image classification. Our goal is to see if modern visual recognition systems (e.g., ConvNext (Liu et al., 2022), ViT Dosovitskiy et al. (2020)) have that human-like behavior to easy/hard-to-classify images. Since there does not exist any real data labeled with ground-truth difficulty, we propose to *generate* one instead. Recent image generative models

have become capable of generating very high quality images (Rombach et al., 2022), good enough to be used in training recognition models (Yu et al., 2023; Azizi et al., 2023), and we believe that they are good enough to be used for our evaluation. With the aid of recent large language and generative models (GPT-4 Achiam et al. (2023) + DALL-E 3 Ramesh et al. (2021)), we design a prompting system to generate descriptions of images of three levels of difficulty. For example, *an image of a fully visible dog* and *an image of dog only partly visible* can be considered to be image descriptions of an easy- and hard-to-classify images respectively. We use DALL-E 3 to take in these different difficulty level prompts and generate images while faithfully preserving the desired attributes. Fig. 1 (left) gives an example.

Once we have the easy, medium and hard-to-classify test images, we record if the model predicts the class correctly or incorrectly. Fig. 1 (right) depicts the 8 possibilities of model's behavior (green/red represent correct/incorrect response). If the model truly learns to classify images by developing the aforementioned notion of easy/hard concepts, then its responses should fall under 4 out of the 8 possibilities highlighted in a red box. Our first key finding is that, for most of the current visual recognition models, their responses *do indeed* fall under the 4 highlighted categories around 80-90% of the time. This result hints that even without an explicit supervision, visual recognition models learn to learn things in a structured way.

While an intriguing result in its own, we believe that this can have applications, especially in the way we evaluate models. We take inspiration from how students are often tested using standardized tests, like the Graduate Record Examination (GRE), for admissions into U.S. universities. These tests are adaptive in nature, where questions in the next round depend on how well the student does in the current one. So, for example, if the student cannot solve easy-medium questions, there is not much point giving them difficult questions in the next round; i.e., one can reliably *predict* that they will get zero points for those hard questions. We develop a similar GRE-type test to evaluate visual recognition models on the generated dataset proposed above. The test is broken up into multiple rounds. In the first round, the model is shown images of medium difficulty on average. Its score in this round determines the distribution of easy/medium/hard questions in the next round. That is, similar to GRE, we can skip over images that are too easy/hard for the model to classify. Thus, instead of evaluating the model on every possible image in the test set, this way of dynamically selecting the images helps approximate that total score of the model on the whole set using only 25% of the test images.

Additionally, the newly proposed dataset can have usefulness in and of itself. We generate images from 100 categories taken from ImageNet (Deng et al., 2009). For each category, we consider 10 attributes. Within each attribute, we generate 12 images for 3 levels of difficulty, bringing the total number of images to 36,000. However, different from standard benchmarks like the ImageNet validation set, these 36k images are labeled with attribute value, difficulty, in addition to the ground-truth class. This can enable analysing models on a much finer level (e.g., ResNet-50 struggles to detect dogs from a side view).

In summary, our work has the following contributions. We present a new method to study the learning dynamics of modern visual recognition systems using the concept of example difficulty. To do this, we create a new test set of synthetic images labeled with class, attribute, and difficulty level. Our results indicate that most of the models do in fact develop a semantically meaningful notion of example difficulty while learning visual concepts, without having access to any external supervision. Using this newly found property, we develop a multi-round adaptive test, inspired by GRE, which steers the future test images according to a model's ongoing performance. This facilitates skipping over too easy/hard questions, and helps assess a model's performance using a fraction of test images.

## 2 RELATED WORK

**Insights into neural network learning mechanisms.** Understanding the internal mechanisms of neural networks is a key focus in deep learning research, with various methods proposed to interpret their generalization, visualize features, and analyze behavior. (i) Generalization vs Memorization: Research shows neural networks can memorize random labels, fueling discussions around this "paradox" (Zhang et al., 2021; Arpit et al., 2017). Studies suggest generalization depends more on training data than model capacity (Dinh et al., 2017; Krueger et al., 2017). (ii) Feature Visualization: Methods like visualizing partial derivatives (Simonyan, 2013) and Class Activation Mapping (CAM) (Zhou et al., 2016) highlight important image regions used by CNNs for decisions. (iii) Model

Behavior: Neural networks often prioritize easy-to-learn features like texture (Geirhos et al., 2018), while harder-to-learn samples, such as shapes, may be neglected (Geirhos et al., 2020). Minority samples in datasets can also be overlooked (Mehrabi et al., 2021). In contrast to these works, our paper studies whether visual recognition models learn concepts in a structured way, where if they can answer something difficult they also can answer something easier, and vice versa, similar to how humans learn. Note that this is related to curriculum learning (Bengio et al., 2009), but the key difference is that in curriculum learning, models are trained on tasks of increasing complexity with explicit or implicit guidance, whereas in our work, we study whether such a capability emerges automatically for standard cross-entropy/contrastive learning-based image classifiers.

**Image classification benchmarks for evaluating fine-grained attributes and difficulty levels.** The Photorealistic Unreal Graphics (PUG) dataset (Bordes et al., 2024), generated using Unreal Engine, provides photorealistic, controllable synthetic data. PUG-ImageNet complements ImageNet with tests for changes in factors like pose, background, size, texture, and lighting. ImageNet-X (Idrissi et al., 2022) offers granular annotations for the ImageNet dataset for naturally occurring attributes such as changes in pose, background, lighting, scale, etc. to identify failure modes, while Spawrious (Lynch et al., 2023) uses image-to-text models to probe classifiers' reliance on spurious correlations. Finally, ImageNet-D (Zhang et al., 2024) uses stable diffusion to create challenging images that test model robustness. To our knowledge, our work is the first to consider both attributes and difficulty levels by designing a prompting system that generates image descriptions across three difficulty levels: Easy, Medium, and Hard, as well as ten different attributes. We then use DALL-E 3 to generate images based on these prompts, ensuring that the specified attributes are accurately represented. This allows for a more detailed analysis of models along both the attribute and difficulty dimensions.

**Real and synthetic benchmarks for robustness evaluation.** The research community has developed several approaches to test model robustness on extended versions of ImageNet (Deng et al., 2009), using datasets that introduce natural distribution shifts (e.g., ImageNetV2 (Recht et al., 2019), ImageNet-Sketch (Wang et al., 2019), ObjectNet (Barbu et al., 2019), ImageNet Rendition (Hendrycks et al., 2021a), ImageNet-Hard (Taesiri et al., 2024), and ImageNet Adversarial (Hendrycks et al., 2021b)) and synthetic shifts (e.g., ImageNet-C (Hendrycks & Dietterich, 2019), ImageNet-9 (Xiao et al., 2020), and Stylized-ImageNet (Geirhos et al., 2018)). However, repeated evaluations on these static benchmarks have leaked cues, reducing their effectiveness (Mayilvahanan et al., 2023). To address this and to reduce the cost of manual collection, recent work generates synthetic images for robustness testing. ImageNet-D (Zhang et al., 2024) uses Stable Diffusion to create challenging images, while Spawrious (Lynch et al., 2023) leverages Stable Diffusion to generate datasets with spurious correlations. PUG (Bordes et al., 2024) offers photorealistic data using Unreal Engine, and AutoEval (Boyeau et al., 2024) combines human and synthetic data for performance evaluation. Our approach stands out by systematically incorporating both difficulty levels (Easy, Medium, Hard) and specific attributes using a detailed prompting system with the powerful generative model DALL-E 3, unlike other benchmarks that focus on either a single difficulty level, have limited attributes (e.g., only foreground and background combinations), or generate unnatural images.

**Adaptive model evaluation.** Inspired by computerized adaptive testing (CAT) (Van der Linden & Glas, 2000), used in exams like the GRE, we develop algorithms for adaptive testing in image classification benchmarks. Similar to CAT, which does not require all questions to be answered and instead efficiently assesses examinees with fewer questions based on examinee responses, our framework evaluates classification models by selecting a subset of samples, reducing computational demands while maintaining assessment accuracy. Prabhu et al. (2024) presents an efficient evaluation framework to maintain a lifelong benchmark by leveraging dynamic programming to rank and sub-select samples, whereas our approach generates unseen images using DALL-E, ensuring models are tested on genuinely new data, mitigating the risk of data leakage, and enhancing the reliability of generalization evaluation.

## 3 VISION MODELS' BEHAVIOR ON EASY → HARD IMAGES

Humans learn concepts in a progressive way - it is only possible for them to solve hard questions if they have the ability to solve easier ones first (Zacks & Tversky, 2001; Newtson, 1973). We want to study if visual recognition models trained for image classification also learn concepts in a similar way. To do this, we first explain in Sec. 3.1 our process of creating a test data having images of various difficulty (akin to questions for humans). Then, in Sec. 3.2, we present a method to analyze a model's response to such images to see if it mimics a human-like way of learning concepts.

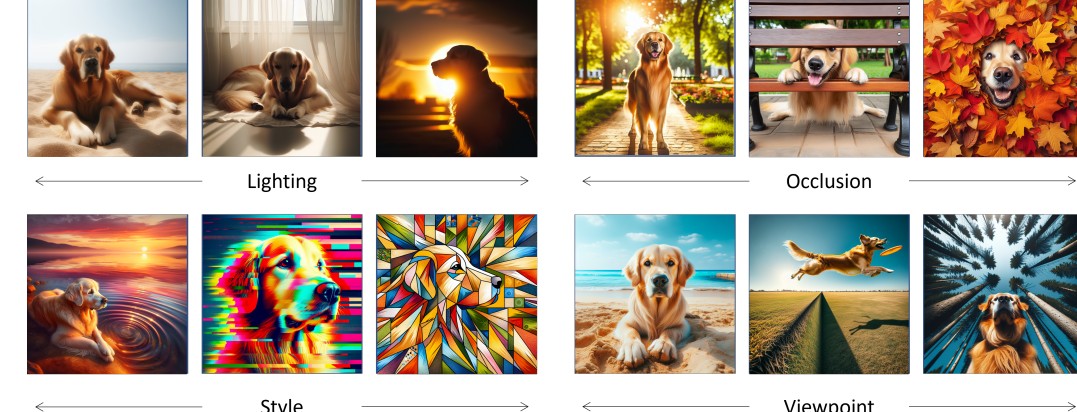

Figure 2: **Visualizing the difficulty of test samples.** All of the images are generated using our proposed pipeline. In each quadrant, we focus on one attribute (e.g., lighting, in the top left), and from left to right we show the images becoming progressively more difficult to be classified correctly.

### 3.1 DATASET CREATION

For the task of image classification, the standard data format is images paired with their ground truth labels $\mathcal{D} = \{(x_1, y_1), (x_2, y_2), ...\}$. However, for our purpose, not only do we need the information that $y_i$ is the ground-truth label of $x_i$, but we also need to know how *difficult* it is for $x_i$ to be classified as $y_i$. Hence, our first task is to formalize this notion of difficulty for our problem.

#### 3.1.1 UNDERSTANDING SAMPLE DIFFICULTY

Consider Fig. 2 and observe the group of three images in the top right corner. All of the images have 'golden retriever' as the main subject. Yet, we, *as humans*, can intuit that it is easiest to correctly classify the content of the left most image and most difficult to do it for the right most one. This is because from left to right, the dog is getting more occluded. Other triplets show similar easy → hard progression for other attributes. The important point from these figures is that the difficulty of a sample is best understood in relation to a particular attribute type (e.g., occlusion). However, to the best of our knowledge, no dataset of real images contains human annotations indicating difficulty about a sample in that way.

Hence, given the recent advances in large language and image generative models, we propose to *synthesize* images with the desired attributes. Text-to-image generative models can now generate very high quality images (Rombach et al., 2022; OpenAI, 2024), so good that they have even been used to train image classification models and shown promise (Yu et al., 2023; Azizi et al., 2023). Furthermore, using the text medium, we can use very precise language to describe more easy/hard-to-classify images. Hence, we propose to use these generative models to design an evaluation setup.

#### 3.1.2 OVERALL IMAGE GENERATION PIPELINE

Since we want to generate images of various difficulty using text, we need the following information for a particular prompt: (i) class name (e.g., golden retriever), (ii) attribute type (e.g., occlusion), and (iii) difficulty type (e.g., hard) for that attribute. Using (ii) and (iii), we can specify a particular attribute value. Here is an example: *An image of a golden retriever heavily occluded by a door*. In this case, *golden retriever* is the image class, *heavily occluded* is the *hard* difficulty for the occlusion attribute. The first step is to collect the names of the classes we wish to evaluate on. We use 100 object categories out of 1000 classes in ImageNet (see appendix A.2 for the complete list). The next step is to generate the attribute values, similar to *heavily occluded* in the above example.

**Generating the attributes.** We first ask GPT-4 (Achiam et al., 2023) to list 10 common attribute types that can help describe image content. The second column of Fig. 3 shows the list. After this, we again prompt GPT-4 with the following for each of those attributes:

```
"To generate text prompts for DALL-E that will generate images
of varying difficulty levels for vision models to classify,
please create nine levels of difficulty based on <attribute name>
```

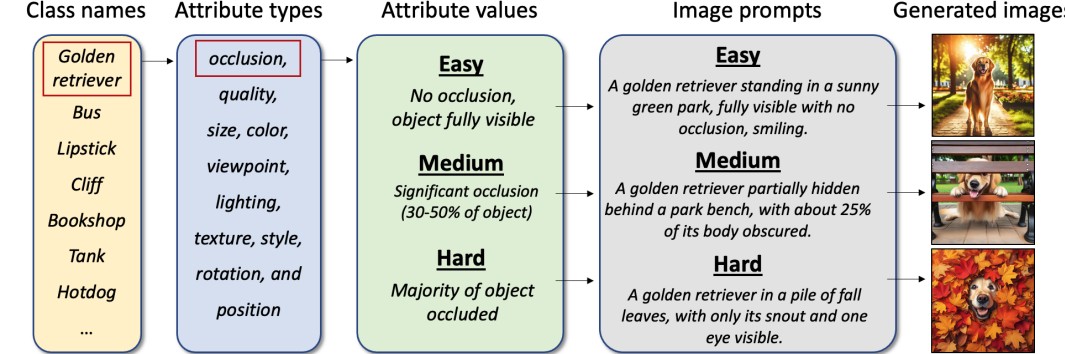

Figure 3: **Overview of the test set generation process.** The first step is to collect the names of the image categories that we wish to test the models on. We then prompt GPT-4 to generate the appropriate attribute values for those categories with various levels of difficulty. Using those, we again prompt GPT-4 to generate text prompts for a category (golden retriever), attribute (heavy occlusion) combination. Finally, we use DALL-E 3 to generate the corresponding images.

attributes and group the nine levels of difficulty into categories
of easy, medium, and hard."

To this, GPT-4 responds with a list of difficulty varying attribute values (descriptions) for each attribute type. As an example, here is how the difficulty of an image can vary along the attribute of *occlusion* - (i) Easy: "No occlusion, object fully visible"; (ii) Medium: "Significant occlusion (30-50% of object)"; (iii) Hard: "Majority of object occluded (70-90%)".

**Generating text prompts.** Given these basic units, we finally prompt GPT-4 one last time to generate the text descriptions for DALL-E 3 to generate the images. This description is produced by combining the information about a certain class with a certain attribute value. For example, *golden retriever* is combined with *heavy occlusion* to produce the prompt - *"A golden retriever in a pile of fall leaves, with only its snout and one eye visible"*. Fig. 3 details the whole process.

**Dataset size:** There are a total of 100 classes, each having the same 10 attributes. For each attribute, there are a total of 3 difficulty levels. And for each difficulty level, there are 12 images. Hence, the total size of our dataset is $10 \times 100 \times 3 \times 12 = 36000$ images. This dataset is balanced across types of class, attribute, and difficulty level.

### 3.2 HIERARCHICAL LEARNING SCORE

We now describe how to use this dataset of easy/medium/hard-to-classify images to study whether models learn concepts hierarchically. If we consider the combination of each class with its attribute type, there will be a total of $100 \times 10 = 1000$ combinations ({*Golden retriever, occlusion*} is one such example). For each such class-attribute combination, we create 12 triplets, each having one easy, one medium, and one hard image. Hence, there are a total of 12000 triplets of test images.

For each visual recognition model that is of interest to us, we test it on these images and record whether their prediction matches the ground-truth object category. The model's response on each triplet will follow one of 8 patterns depending on whether it gets the easy/medium/hard image correctly or incorrectly classified; these are shown in Fig. 1 right (red/green colors mean correct/incorrect prediction respectively). We compute the percentage of each of these 8 patterns over all 12,000 triplets. Now, if we consider the following human-like hierarchical learning principle - *A model should correctly answer a harder question only if it can answer all easier questions*, we see that only 4 out of those 8 patterns satisfy the requirement (highlighted with red box). We call them 'principle-following patterns', and define the **hierarchical-learning score** to be the *percentage of model's responses on 12000 triplets that fall under principle-following patterns*. The more a model follows this principle, the higher its hierarchical-learning score should be.

**Experiments and Results:** We choose three popular architectures - (i) ViT-B16 (Dosovitskiy et al., 2020), (ii) ConvNext (Liu et al., 2022) and (iii) ResNet-101 (He et al., 2016) - each trained on ImageNet1k (Deng et al., 2009) and LAION (Schuhmann et al., 2022) using cross-entropy and CLIP objective (Radford et al., 2021), respectively. This gives us a total of 6 trained models which we test using the setting described above. The results are shown in Fig. 4 (left), where we plot the percentage

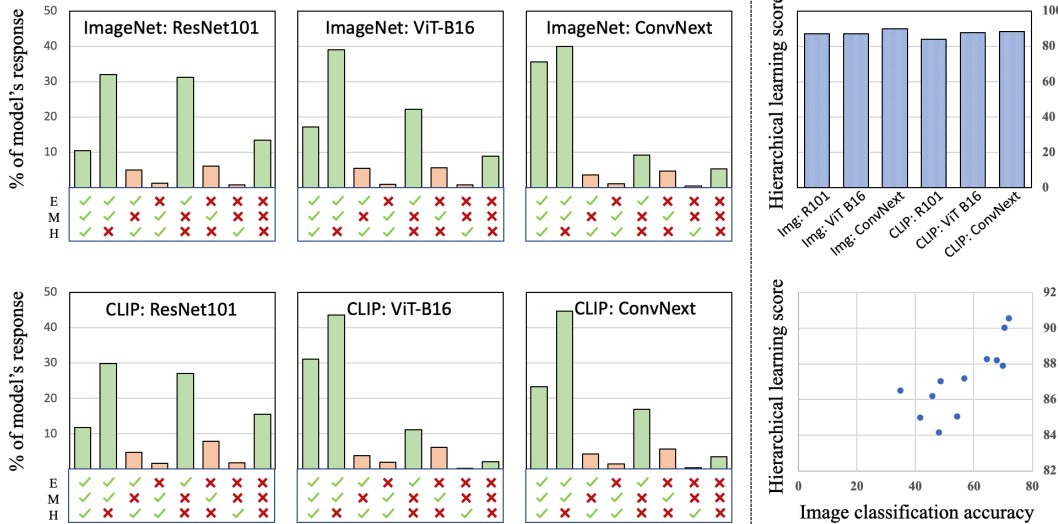

Figure 4: **Left:** Plots depicting % of model's behavior on 12k triplets over the 8 possible patterns of **E**asy, **M**edium, **H**ard. The bars corresponding to principle-following pattern are colored green; others, red. All models behave according to the hierarchical learning principle. **Top right:** Hierarchical learning score of 6 vision models. Most achieve a score higher than 85%. **Bottom Right:** Scatter plot of top-1 accuracy on our test set vs hierarchical learning score of 12 models. PCC value is 0.77.

of a model's behavior on 12000 triplets over all the patterns. We color the bars corresponding to principle-following and not following patterns as green and red respectively. Deriving from these plots, we also report the hierarchical-learning score of these 6 models in Fig. 4 (top right). First, for all models, the majority behavior on triplets falls under the principle-following pattern, and almost all get a high hierarchical-learning score of >85%. In fact, in all cases, the 4 most frequent behaviors correspond to those 4 principle-following patterns. The most frequent behavior across all models is {Easy: ✓, Medium: ✓, Hard: × }, while the least frequent one is {Easy: ×, Medium: ×, Hard: ✓ }. Also, whether a model was trained on ImageNet using cross entropy loss or on LAION to align images and text, e.g., ResNet101 vs CLIP:ResNet101, the behavior remains roughly the same. These results indicate that common visual recognition models do in fact follow the human-like hierarchical learning principle, even when there is no explicit supervision to do so.

**Model's hierarchical learning score vs accuracy:** Consider two extreme scenarios. Models A and B are so good and bad at classifying images that most of their behavior is of the form {✓, ✓, ✓} and {×, ×, ×} respectively. Even though their top-1 classification accuracies could be vastly different, they would both still have a very high hierarchical-learning score. So, it is unclear what, if anything, the hierarchical-learning score has to do with classification accuracy. We empirically study this, by collecting a total of 12 models (6 additional ones compared to previous study; see appendix) and for each, compute its hierarchical-learning score and top-1 accuracy on our test data. The scatter plot in Fig. 4 (right) shows these data points. There is a correlation between the top-1 accuracy and the hierarchical learning score; the Pearson correlation coefficient is 0.77. Since this is merely a correlation, it is difficult to say whether more accurate models become more accurate *because* they learn to learn concepts in a human-like style. Nevertheless, a positive correlation is a sign that, contrary to the extreme example we discussed above, the learning dynamics of very incapable and capable models is not symmetric. However, we point out that even the least hierarchical-learning score for a model is 84.2, which, for the purposes of this work, we consider high enough to say that the corresponding model still follows the hierarchical-learning principle.

## 4 ADAPTIVE TESTING OF IMAGE CLASSIFIERS

The fact that humans learn concepts in a hierarchical way has had applications in the way their learning gets tested. Many standardized tests like the GRE are adaptive in nature. Based on how well the student is doing at any point, future questions are adapted to match the student's ability. For example, if a student is struggling to answer questions like $(2 \times 3)$ and $(4 \times 5)$, there is not much point in testing them on $(2^2 + 3^2) \times (4 + 5)$. We can reasonably predict that they will get zero points for the latter question, and hence can avoid testing them on all possible (easy/medium/hard)

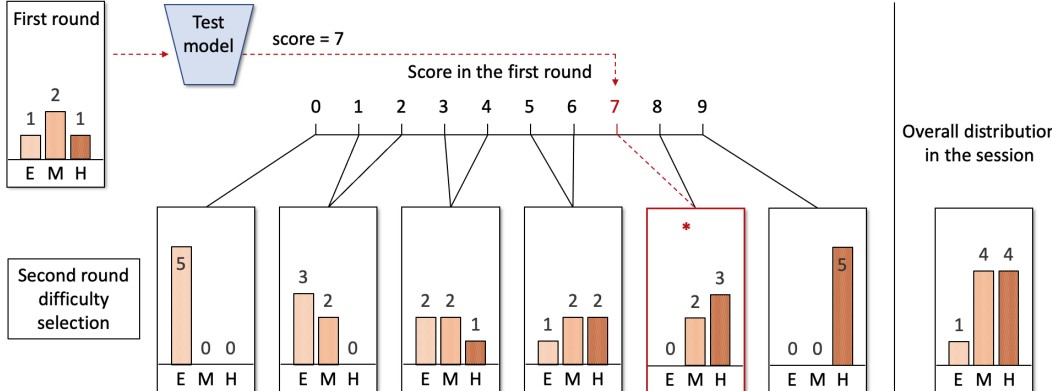

Figure 5: **Adaptive testing of a classifier.** The test involves two rounds. Similar to GRE, the first round is of, on average, medium level difficulty, consisting of 4 test images (1 easy, 2 medium, 1 hard). The model gets a score (max = 9, min = 0) based on which the distribution of images for the next round is chosen. We show an example of a model getting a score of 7 in round 1, because of which in next round there are 0 easy, 2 medium, and 3 hard images. **Right:** Throughout the whole session, the model gets tested on a total of 9 images; in this case, 1 easy, 4 medium and 4 hard images.

questions, while still being able to accurately assess a student's capability. Similarly, our setting also has a test set of 36k images uniformly spread over 100 object categories, 10 attribute types, and 3 levels of difficulty. Given that visual recognition models also seem to learn in a hierarchical way, we investigate a similar GRE-type test for them, using which we can get a good estimate of model's overall performance without needing to test it on every single easy/medium/hard image.

To test a model, we start by iterating over all the classes and attribute types. In each such class + attribute type combination (there are 1000 of these), there are 3 difficulty levels, each with 12 images, i.e., $3 \times 12 = 36$ test images. The goal is to predict how a model performs on these 36 images without needing to evaluate it on all 36 images. Hence, we conduct a test borrowing principles from GRE. Instead of the model being tested on all images in the same round at once, we consider two rounds of test images. In the first round, we randomly select 1 easy, 2 medium, and 1 hard image. The average difficulty is medium. The model is tested on these 4 images, and we assign it a score based on how well it did. Similar to GRE, one gets more points for solving harder images. We assign the model 1, 2, and 4 points for correctly classifying an easy, medium, and hard image, respectively; and 0 points for misclassification. In the next round, the model is tested on 5 new images. The distribution of these new images depends on the model's total score in the first round (min/max score is 0/9 respectively). The mapping from total score to second round images' distribution is shown in Fig. 5. Similar to the first round, the model gets assigned points for correctly classifying images. All throughout the process, the model is tested only on $x$ easy, $y$ medium, and $z$ hard test images, where $x + y + z = 9$. So, after the end of the second round, we can collect the following information about the model: (i) its total score, and (ii) classification accuracies on easy/medium/hard images.

Once the model has gone through all the class/attribute combinations, and we have collected these information from each session (first + second round), there are multiple ways to combine that information to analyze the model. We can get an attribute level score by averaging each session's score across all the 100 classes. Similarly, we can get attribute level accuracy, either overall across easy/medium/hard questions or per difficulty, also averaged across the classes. This is helpful if we want to study which attributes a model struggles at, discussed more in Sec. 5. Obviously, one can get a global score/accuracy, averaged across attributes dimension, for the model. The key part in all of these analyses is that they can be done using only a fraction of the 36k images. In the next section, we discuss how accurate our predictions (score/accuracy) can be when compared against those same values computed over the whole set.

## 5 EXPERIMENTS

In this section, we first validate whether the generated images are appropriately categorized into easy, medium, and hard difficulty levels. We then evaluate the prediction results of adaptive testing against full dataset counterpart results. Lastly, we provide a detailed error analysis of various image classifiers for different attributes and difficulty levels.

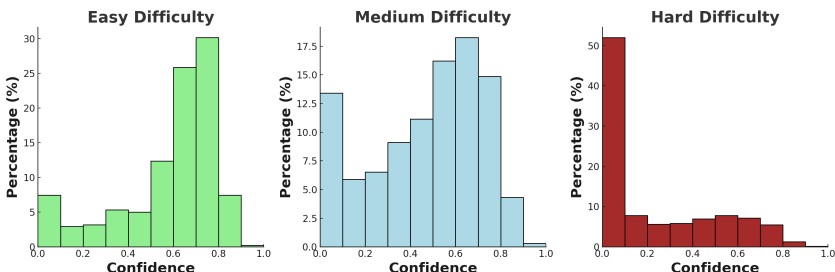

Figure 6: % of our dataset grouped according to classification confidence for the Easy, Medium, and Hard difficulty levels. We average the sample numbers across six selected classifiers (ViT-B16 (Dosovitskiy et al., 2020), ConvNext (Liu et al., 2022), ResNet-101 (He et al., 2016), trained on ImageNet1k (Deng et al., 2009) and LAION (Schuhmann et al., 2022)).

Table 1: Average number of questions tested per difficulty level for adaptive testing. Models with better performance tend to receive a higher proportion of medium and hard questions, and vice versa.

| Difficulty | Easy | Medium | Hard |
|---|---|---|---|
| ResNet18 | 3.51 | 3.83 | 1.55 |
| ResNet101 | 2.56 | 3.96 | 2.48 |
| ViT-B16 | 2.0 | 3.89 | 3.07 |
| ConvNext-B | 1.55 | 3.46 | 3.98 |
| CLIP-RN101 | 2.60 | 3.89 | 2.51 |
| CLIP-ViT-B16 | 1.47 | 3.61 | 3.93 |
| CLIP-ConvNext-B | 1.62 | 3.79 | 3.58 |

## 5.1 EVALUATING CORRECTNESS OF DIFFICULTY LEVELS IN GENERATED IMAGES

A key ingredient to test our hypothesis, i.e., whether vision models follow the hierarchical learning principle, has been the usage of generated images. The images shown in Fig. 2 do look good to our eyes and seem to make sense (some are easier to classify, others more difficult). However, to make sure that our dataset as a whole is meaningful and does not contain garbage images, we verify whether those same images of various difficulty can be appropriately categorized into their respective difficulty levels. To accomplish this, we use six classifiers to assess the prediction confidence for each sample at each difficulty level. For the easy difficulty level, we expect a high proportion of samples to exhibit high prediction confidence. Conversely, for the hard difficulty level, we anticipate a significant number of samples to display low prediction confidence. For the medium difficulty level, we expect the prediction confidence of many samples to fall between the easy and hard difficulty levels. By analyzing the distribution of prediction confidence across the difficulty levels for our entire dataset of 36,000 images in Figure 6, we validate the efficacy of our image generation process and ensure that the generated images accurately represent the intended difficulty levels.

## 5.2 IS THE TEST DIFFERENT FOR DIFFERENT MODELS?

The purpose of the adaptive test introduced in Sec. 4 was to test more capable models on harder samples, and not-so-capable models on easier ones. Given that in each session (first + second rounds) we test any model on a total of 9 questions, we should hence expect some diversity in what those 9 questions look, on average, for different models. In Table 1, we plot the average distribution of each/medium/hard test images that 7 visual recognition models face in a session. We see that most models get tested the most on medium questions. However, there are significant differences among certain models. For example, ResNet18, which is the weakest model in the list (ImageNet validation top-1 accuracy = 69.76%) gets tested much more on easier images (3.51) than hard ones (1.55). In contrast, a stronger & capable model like ConvNext (ImageNet validation top-1 accuracy = 84.06%) gets tested much more on harder (3.98) than easier images (1.55). So, these results tells us how different models, owing to their differing capabilities, create their own unique trajectories of test questions that they get evaluated on.

Table 2: We evaluate each classifier's score and accuracy using Static 3 and our adaptative method, comparing their 10-dimensional vectors—each element representing the score or accuracy of a specific attribute—against Static 12 using the Mean Squared Error. The evaluation is repeated three times, and the average error is reported. Our method provides accurate performance estimates with fewer test images and smaller errors than Static 3, optimizing the evaluation process. (Static #) represents the number of test images used in each level of difficulty for each attribute of a given class.

|  | Classifier | RN101 | ViT-B16 | ConvN-B | C-RN101 | C-ViT-B16 | C-ConvN-B |
|---|---|---|---|---|---|---|---|
| Score | Static 12 | 35.3 | 43.8 | 59.2 | 36.4 | 57.6 | 51.0 |
| Score Error | Static 6 | 0.8 | 0.7 | 0.8 | 0.9 | 1.1 | 1.1 |
| Score Error | Static 3 | 5.4 | 4.6 | 3.2 | 4.5 | 5.5 | 4.8 |
| Score Error | Ours | 4.2 | 2.7 | 2.5 | 2.2 | 1.5 | 3.3 |
| Acc | Static 12 | 48.5 | 56.9 | 70.2 | 48.1 | 69.8 | 64.6 |
| Acc Error | Static 6 | 0.6 | 0.6 | 0.5 | 1.0 | 0.7 | 0.6 |
| Acc Error | Static 3 | 4.9 | 2.6 | 2.0 | 4.4 | 3.5 | 1.6 |
| Acc Error | Ours | 3.6 | 1.0 | 0.9 | 3.6 | 2.3 | 0.9 |

## 5.3 How closely does Adaptive Evaluation follow Full Evaluation?

We next evaluate our GRE-style adaptive evaluation discussed in Sec. 4, comparing to what the performance would be with the standard way of evaluating a model on the entire dataset across the ten attributes and three difficulty levels.

To measure classification performance of a model, in addition to standard classification accuracy, we compute a GRE-style score to reward getting more difficult examples correct: Score = (correct$_{easy}$ × 1) + (correct$_{medium}$ × 2) + (correct$_{hard}$ × 4), i.e., 1, 2, and 4 points for correctly classifying an easy, medium, and hard image; and 0 points for misclassification. Note, one advantage of using a GRE-style score is its ability to differentiate between models with similar accuracy. For example, when two models, such as ConvNext-B and CLIP-ViT-B16, achieve close accuracy scores (70.2 and 69.8, respectively) in Table 2, the GRE-style score (59.2 vs. 57.6) can highlight the differences by showing which model is more successful in handling more challenging questions.

Our full generated test set has a total of 36,000 images—comprising 12 images for each combination of 100 classes, 10 attributes, and 3 difficulty levels. We refer to the evaluation results from this complete set as 'Static 12,' which serves as our ground truth (full evaluation baseline). To validate our adaptive testing method, we first establish a baseline called 'Static 3.' This baseline is created by randomly selecting 3 images out of the 12 for each difficulty level. This produces a subset containing 100 (classes) × 10 (attributes) × 3 (difficulty levels) × 3 (images) = 9,000 images. Our adaptive testing then selects 9 samples across the three difficulty levels for each attribute and class, allowing the number of selected images for each difficulty level to vary. This also results in a subset of 100 (classes) × 10 (attributes) × 9 (images) = 9,000 images, matching the size of the Static 3 subset, but with a different difficulty distribution than that of Static 3. We then evaluate the classifier on these subsets, aiming to achieve results that correlate with the full test set ('Static 12') while using fewer images and potentially maintaining smaller errors compared to the Static 3 strategy.

We compare the performance of Static 3 and our adaptive testing with the full evaluation (Static 12) in Table 2. This evaluation is repeated three times, and the average error is reported. The results indicate that our adaptive testing provides accurate performance detail estimates using fewer test images than the full set and achieves smaller score and accuracy errors compared to the Static 3 strategy. This confirms that based on the hierarchical learning pattern of common image classification models, it is possible to perform adaptive testing to expedite the evaluation process.

## 5.4 Detailed Error Analysis

Since our generated dataset provides granular labels for attributes such as size, color, lighting, occlusions, and style, we can identify specific failure modes in model performance. We analyze the mistake types of the six classifiers in Table 3. Similar to what was observed in the ImageNet validation set (Idrissi et al., 2022), models with similar overall scores tend to have similar per-attribute scores. For instance, ConvNext-Base and CLIP ViT B16 not only share similar overall scores (59.2

Table 3: Score of different models for each attribute. Bold/underline indicates best/second best.

| Attributes | Color | Light | Occlu | Pos | Quality | Rot | Size | Style | Texture | View |
|---|---|---|---|---|---|---|---|---|---|---|
| ResNet101 | 41.1 | 31.1 | 40.2 | 39.1 | 47.0 | 34.8 | 30.4 | 23.7 | 30.4 | 35.2 |
| ViT-B16 | 47.7 | 39.7 | 46.6 | 48.9 | 56.9 | 48.0 | 36.3 | 34.2 | 37.8 | 41.6 |
| ConvNext-B | **63.7** | **60.6** | **59.4** | **86.7** | 66.2 | **77.5** | **40.8** | 41.3 | 42.9 | **52.6** |
| CLIP-RN101 | 39.5 | 33.4 | 33.3 | 62.2 | 36.6 | 39.7 | 34.6 | 30.2 | 23.4 | 31.2 |
| CLIP-ViT-B16 | 62.2 | 58.9 | 46.6 | 85.3 | **67.2** | 59.8 | 39.8 | **53.3** | **50.4** | 52.4 |
| CLIP-ConvNext-B | 53.1 | 49.9 | 48.4 | 79.4 | 57.4 | 47.0 | 36.7 | 45.1 | 41.1 | 47.7 |

vs 57.6) but also have 6 out of 10 very close attribute scores. Most models, even as their overall scores improve, consistently struggle with certain factors like size, texture, style, and viewpoint. Conversely, they perform well on factors such as object position and image quality. In addition to analyzing attribute-level errors, our generated dataset enables a detailed difficulty-level analysis for each classifier, as shown in Tables 5, Table 6, and Table 7. Across all models, the performance decreases as the difficulty level increases. Attributes like 'Texture', 'Style', and 'Viewpoint' generally have lower accuracies, especially at the 'Hard' level. One special case is for the 'size' attribute; all six models perform generally well on easy and medium difficulty but face significant challenges at a hard level, which often includes examples with many tiny objects.

## 6 DISCUSSION AND LIMITATIONS

The typical way one evaluates a visual recognition model is to compute its average top-1 accuracy over a standard test set (e.g., ImageNet). However, there have been concerns that models can get over optimized on such benchmarks (Recht et al., 2019). We believe that our work can help in a better assessment of models in two ways. First, since we can always generate (on-the-fly) a new fresh test dataset for evaluation, there is much less chance for a model overfitting, even if one is simply interested in classification accuracy. Second, the hierarchical learning score measures a different dimension of learning ability; it is less about how accurate a model is, but more about whether that capability is grounded in some fundamental learning principle, or in memorizing examples.

However, this score is not immune to extreme cases. If the model to be tested has completely overfitted to test images, then it can get $\{\checkmark, \checkmark, \checkmark\}$ for most of them and hence get a high hierarchical learning score. Finally, while our approach of using DALL-E 3 to generate a synthetic dataset offers a novel way to study model behavior across varying difficulty levels, it has certain limitations. DALL-E sometimes struggles with generating images for unfamiliar classes, such as "African Hunting Dog," often producing images of other dog breeds labeled incorrectly, which can lead to noisy annotations. Additionally, for complex or uncommon prompts like "A carousel in an amusement park, almost entirely hidden behind a festival tent," DALL-E may generate simplified images, such as a festival tent near a carousel but without the intended occlusion, resulting in unintended "easy" sample. These issues could introduce bias into our dataset, potentially affecting the balanced evaluation of models. We manually reviewed and removed some of these examples, but future work could explore more advanced generative models to address these challenges.

## 7 CONCLUSION

In this work, we explored whether modern visual recognition models exhibit human-like learning behaviors when classifying images of varying difficulty. By leveraging advanced generative models (GPT-4 and DALL-E 3), we generated a synthetic dataset annotated with class, attribute, and difficulty level to test this concept. Our findings reveal that most models do, in fact, demonstrate a structured understanding of example difficulty, even without explicit supervision. This insight opens up new possibilities for model evaluation, leading us to develop a multi-round adaptive test inspired by standardized exams like the GRE. This adaptive testing method significantly reduces evaluation time by selecting test images based on the model's ongoing performance, providing a comprehensive assessment using only a subset of the dataset. Additionally, the synthetic dataset itself, with its detailed annotations, offers a valuable resource for more granular model analysis. Our work contributes a novel perspective on understanding learning dynamics in visual recognition models and proposes an efficient, dynamic approach to their evaluation.

## 8 REPRODUCIBILITY STATEMENT

The results and figures in this paper will be reproducible using our open-source code and the synthetic benchmarks we generated. We will release the synthetic benchmark, with each sample annotated for both attribute and difficulty level.

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
