## A APPENDIX

### A.1 ATTRIBUTE OF VARIATION DEFINITIONS

| Attribute of variation | Description |
|---|---|
| Position | The location or placement of the object within the frame of the image. It can indicate whether the object is centered, towards the edge, or even partially out of view. |
| Viewpoint | Describes the angle or perspective from which the object is observed, such as front, side, top-down, or oblique view. The viewpoint affects the amount of detail visible and can reveal or obscure specific features of the object. |
| Quality | Indicates the overall clarity and resolution of the image. High-quality images have fine details and little noise, while low-quality images may appear blurry, pixelated, or noisy, making it harder to discern specific features. |
| Rotate | Describes the orientation of the object in the image. An object can be upright, tilted, or flipped. The rotation can affect the perception and recognition of the object's standard appearance. |
| Occlusion | Occurs when parts of the main object are blocked or obscured by other objects in the scene. This can make it challenging to identify the full structure of the object. |
| Size | Refers to the object's scale within the image. Size can be influenced by the object's actual size, its distance from the camera, or the zoom level. |
| Lighting | Lighting in the image is either brighter or darker when compared to the prototypical images. |
| Color | Color can indicate the object's natural appearance, the time of day, or the overall mood. |
| Texture | Refers to the surface quality or pattern seen on the object, such as smooth, rough, glossy, or matte. |
| Style | Indicates the visual aesthetics or artistic rendering of the image. This could include photographic styles (e.g., realistic, abstract, cartoonish), drawing styles, or filters applied to the image. |

### A.2 LIST OF 100 OBJECT CATEGORIES

We selected 100 object categories from the 1,000 classes in ImageNet for our study. These categories represent a diverse range of items, animals, and objects, including: Objects: catamaran, wooden spoon, hourglass, stopwatch, iPod, plate, crate, turnstile, frying pan, comic book, pencil box, cash machine, school bus, obelisk, volleyball, lifeboat, computer keyboard, CD player. Animals: malamute, koala, goose, meerkat, gazelle, bullfrog, loggerhead turtle, box turtle, iguana, Komodo dragon, rock python, diamondback rattlesnake, scorpion, wolf spider, black grouse, flamingo, king penguin, killer whale, Chihuahua, Maltese dog, beagle, Afghan hound, Irish wolfhound, Border collie, Rottweiler, Bernese mountain dog, Dalmatian, Siberian husky, lion, tiger, American black bear, ladybug, fire salamander, hummingbird, goldfinch, toucan, peacock, lobster, Dungeness crab, zebra, bison, hippopotamus, giraffe, kangaroo, platypus, woodpecker, raccoon, skunk, bat, otter, seahorse, jellyfish, sea anemone, coral, stork, crane, tortoise, parrot. Food-related: beer bottle, lipstick, mixing bowl, mashed potato. Others: cliff, black widow, lakeside, sock, great white shark, ostrich, bald eagle, vulture, American

alligator, African elephant, golden retriever. This wide range of categories ensures a comprehensive evaluation of model performance across various domains.

### A.3 VISUALIZING THE DIFFICULTY OF TEST SAMPLES

We present additional images featuring a golden retriever as the main subject, focusing on attributes such as color, texture, quality, and size. From left to right, the images are arranged to become progressively more challenging for accurate classification. Please see Fig. 7. Finally, we also show more examples for other classes along with their attributes in Fig. 8, 9, 10, 11.

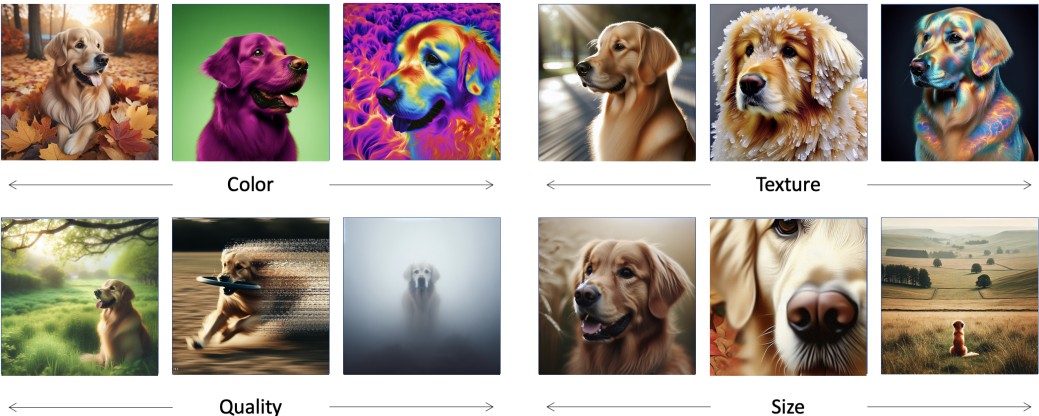

Figure 7: **Visualizing the difficulty of test samples.** All of the images are generated using our proposed pipeline. In each quadrant, we focus on one attribute (e.g., color, in the top left), and from left to right we show the images becoming progressively more difficult to be classified correctly.

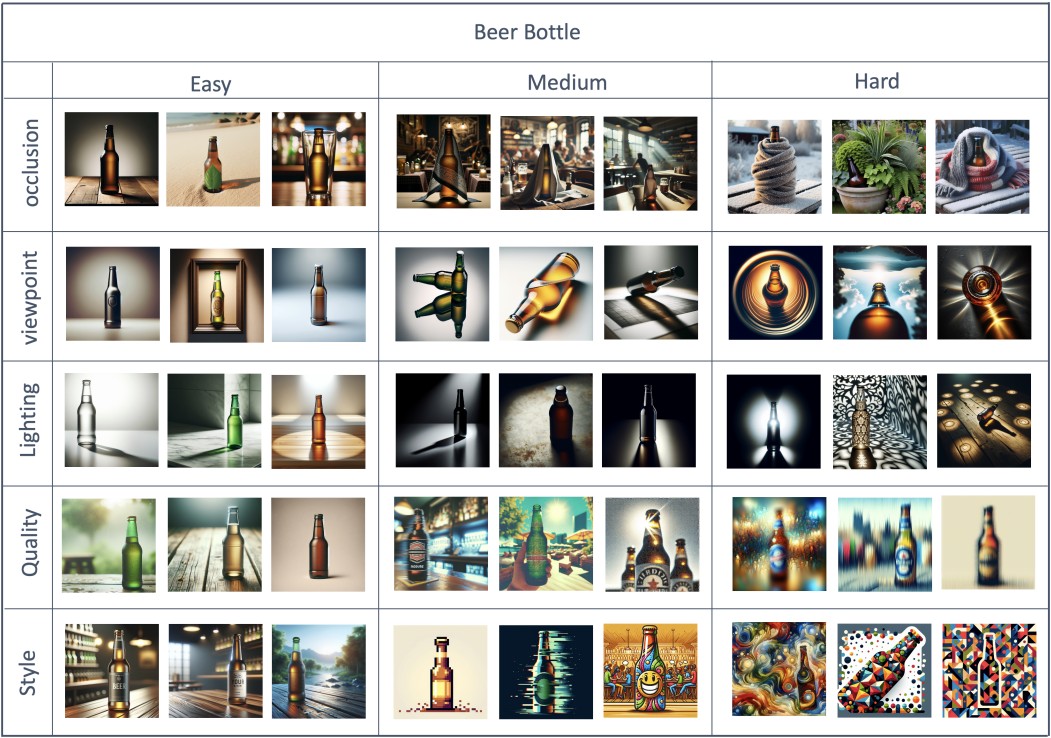

Figure 8: **Visualizing the class of Beer Bottle.**

Figure 9: **Visualizing the class of African elephant.**

## A.4 DETAILED ERROR ANALYSIS

In addition to analyzing attribute-level errors, our generated dataset enables a detailed difficulty-level analysis for each classifier, as shown in Tables 5, Table 6, and Table 7. Across all models, the performance decreases as the difficulty level increases. This is a general trend for each attribute, indicating that all models struggle more with "Hard" samples compared to "Easy" and "Medium" ones. Additionally, attributes like "Texture," "Style," and "Viewpoint" generally have lower accuracies, especially at the "Hard" level. This suggests that these attributes pose more significant challenges for current deep-learning models.

| Attribute | CLIP ResNet101 | ResNet101 | CLIP ViT B16 | ViT B16 | CLIP ConvNext Base | ConvNext Base | Average (Attributes) |
|---|---|---|---|---|---|---|---|
| Color | 58.89 | 70.74 | 83.33 | 75.56 | **84.07** | 83.70 | 76.38 |
| Lighting | 67.04 | 67.04 | **91.11** | 77.41 | 87.41 | 82.59 | 78.77 |
| Occlusion | 65.93 | 76.67 | 84.81 | 80.00 | **88.52** | 86.67 | 80.77 |
| Position | 97.78 | 96.67 | **100.00** | 97.04 | 99.26 | 97.04 | 97.96 |
| Quality | 69.26 | 78.52 | **89.26** | 80.74 | 87.41 | 87.41 | 82.77 |
| Rotate | 99.26 | 96.67 | **100.00** | 97.78 | **100.00** | 99.26 | 98.49 |
| Size | 98.52 | 97.04 | **100.00** | 98.15 | **100.00** | 99.26 | 98.83 |
| Style | 71.48 | 68.89 | 82.96 | 78.52 | **85.56** | 82.22 | 78.27 |
| Texture | 42.96 | 56.67 | **77.78** | 67.04 | 75.19 | 75.19 | 65.64 |
| Viewpoint | 63.70 | 77.41 | 86.67 | 84.81 | 84.44 | **89.63** | 81.11 |
| **Average** | 73.08 | 77.13 | **89.59** | 83.00 | 89.19 | 88.30 | |

Table 5: Accuracy for different attributes at the easy difficulty level. Bold indicates the highest score, and underline denotes the second highest. The rightmost column shows the average accuracy of each attribute.

## A.5 HIERARCHICAL LEARNING SCORE OF ADDITIONAL MODELS

As Section 3.2 mentions Hierarchical Learning Score (HLS), we include an additional six classifiers: ResNet 18, ResNet 50, ConvNext Large, ConvNext Small, ViT Small 16, and ViT Large 16. Their Hierarchical Learning Scores are provided in Table 8.

Figure 10: **Visualizing the class of Koala.**

| Attribute | CLIP ResNet101 | ResNet101 | CLIP ViT B16 | ViT B16 | CLIP ConvNext Base | ConvNext Base | Average (Attributes) |
|---|---|---|---|---|---|---|---|
| Color | 50.37 | 51.48 | 78.89 | 66.29 | 69.63 | **81.85** | 66.42 |
| Lighting | 48.52 | 47.78 | **84.44** | 55.93 | 75.19 | 80.37 | 65.71 |
| Occlusion | 47.41 | 57.78 | 72.59 | 62.96 | 71.48 | **80.00** | 65.37 |
| Position | 67.41 | 38.89 | 93.70 | 54.44 | 91.11 | **94.81** | 73.73 |
| Quality | 43.70 | 60.74 | 78.89 | 67.78 | 75.19 | **77.04** | 67.22 |
| Rotate | 56.67 | 44.44 | 94.07 | 69.63 | 75.19 | **96.30** | 72.05 |
| Size | 62.22 | 54.07 | 81.85 | 70.74 | **85.19** | 85.19 | 73.54 |
| Style | 49.26 | 35.19 | **84.44** | 56.67 | 78.52 | 66.29 | 61.06 |
| Texture | 40.37 | 49.26 | **78.89** | 57.41 | 69.26 | 68.52 | 60.62 |
| Viewpoint | 44.07 | 56.29 | 80.74 | 65.56 | 67.78 | **82.96** | 66.23 |
| **Average** | 50.40 | 49.69 | **82.85** | 62.44 | 75.65 | 81.03 | |

Table 6: Accuracy for different attributes at the medium difficulty level. Bold indicates the highest score, and underline denotes the second highest. The rightmost column shows the average accuracy of each attribute.

## A.6    MORE CONFIDENCE VISUALIZATION FOR THE EASY, MEDIUM, AND HARD DIFFICULTY

In this section, we visualize the distribution of prediction confidence across the difficulty levels for several classifiers, using our generated dataset. Please see Fig. 12 and 13. We see that they follow a similar trend as described in Fig. 6, where the distribution of confidence is progressively decreasing as we move from easy → hard samples.

## A.7    IMAGE GENERATION PIPELINE

Please see Fig. 14 for the detailed view of all the prompts used to create the final text caption used by DALLE-3 to generated the images.

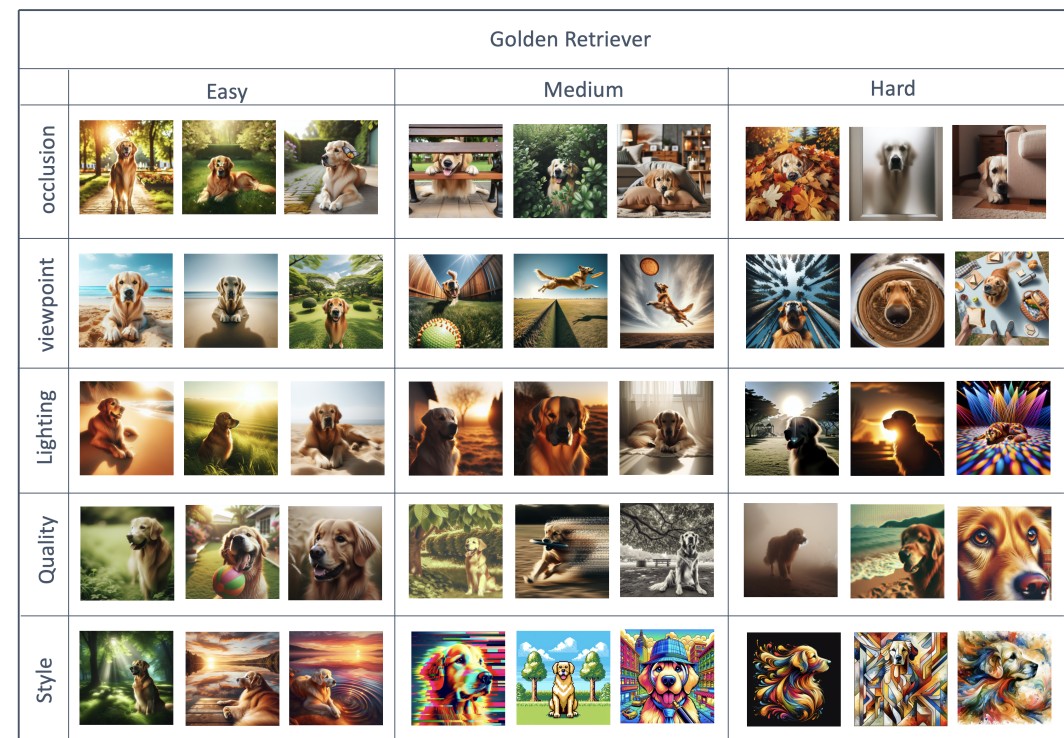

Figure 11: **Visualizing the class of golden retriever.**

| Attribute | CLIP ResNet101 | ResNet101 | CLIP ViT B16 | ViT B16 | CLIP ConvNext Base | ConvNext Base | Average (Attributes) |
|---|---|---|---|---|---|---|---|
| Color | 29.26 | 28.52 | _48.52_ | 31.48 | 37.04 | **47.04** | 36.98 |
| Lighting | 17.41 | 13.70 | _38.15_ | 22.22 | 30.74 | **45.19** | 27.57 |
| Occlusion | 18.15 | 22.22 | 24.07 | _30.00_ | 26.67 | **44.07** | 27.53 |
| Position | 50.74 | 38.89 | _77.41_ | 34.07 | 68.15 | **80.37** | 58.27 |
| Quality | 24.81 | 32.22 | _55.93_ | 45.56 | 43.33 | **52.59** | 42.07 |
| Rotate | 16.30 | 14.44 | _32.59_ | 24.81 | 19.63 | **62.59** | 28.06 |
| Size | 4.81 | 1.85 | _3.70_ | 3.70 | 1.85 | **4.44** | 3.39 |
| Style | 10.37 | 6.67 | _30.37_ | 11.85 | 20.37 | **21.48** | 16.52 |
| Texture | 10.00 | 14.44 | _29.26_ | 20.74 | 18.52 | **22.22** | 19.20 |
| Viewpoint | 16.67 | 14.07 | **29.63** | 18.89 | _28.51_ | 28.15 | 22.26 |
| **Average** | 19.65 | 18.60 | _36.36_ | 26.83 | 29.75 | **38.82** | |

Table 7: Accuracy for different attributes at the hard difficulty level. Bold indicates the highest score, and underline denotes the second highest. The rightmost column shows the average accuracy of each attribute.

Table 8: Hierarchical Learning Score of additional six visual recognition models.

| Classifer | ResNet18 | ResNet50 | ConvNext-L | ConvNext-S | ViT-S16 | ViT-L16 |
|---|---|---|---|---|---|---|
| HLS | 86.52 | 86.19 | 90.56 | 88.22 | 85.00 | 85.04 |

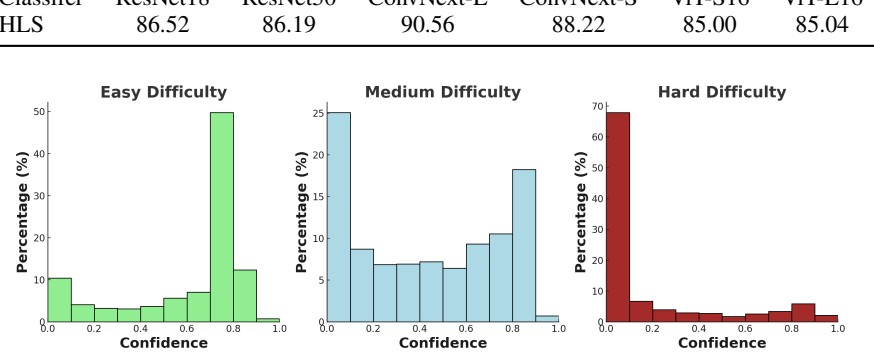

Figure 12: **classification confidence for ViT-B16 model.**

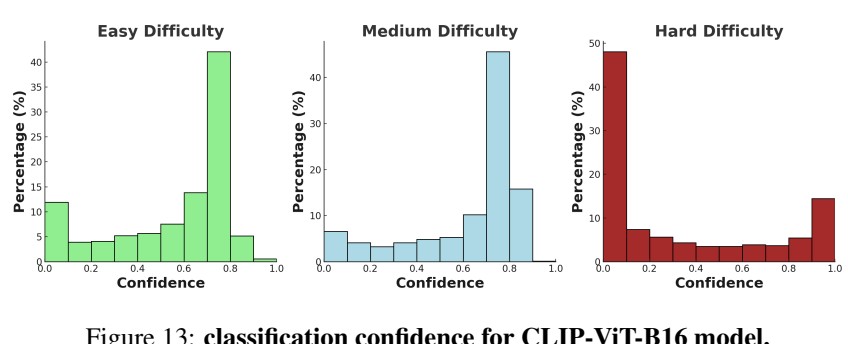

Figure 13: **classification confidence for CLIP-ViT-B16 model.**

Please generate ten distinct attributes of images, which should cover a range of visual characteristics commonly used in image analysis

attribute name

To generate text prompts for DALL-E that will generate images of varying difficulty levels for vision models to classify, please create nine levels of difficulty based on <attribute name> attributes and group the nine levels of difficulty into categories of easy, medium, and hard.

E/M/H attribute
value

Please create text prompts for DALL-E to generate images of <specified class> class, each varying in <attribute name> attribute that progressively increases in difficulty for a vision model to classify. I will provide three levels of difficulty, with each level including multiple examples for guidance. Here are the details:

**Easy:**
-   Easy difficulty attribute value 1
-   Easy difficulty attribute value 2
-   Easy difficulty attribute value 3
**Medium:**
-   Medium difficulty attribute value 1
-   Medium difficulty attribute value 2
-   Medium difficulty attribute value 3
**Hard:**
-   Hard difficulty attribute value 1
-   Hard difficulty attribute value 2
-   Hard difficulty attribute value 3

For each difficulty level, generate three distinct text prompts accordingly.

Text prompts

Generate an image based on the following description: <text prompts>

Figure 14: **Image generation Pipeline.**