# OpenReview forum: "Do Vision Models Develop Human-Like Progressive Difficulty Understanding?"
_ICLR.cc/2025/Conference — Submitted to ICLR 2025_

### Official Review · Reviewer_1Azw · 2024-10-30

**Soundness:** 3
**Presentation:** 4
**Contribution:** 3
**Rating:** 5
**Confidence:** 4

**Summary:**

This paper is about evaluating a new kind of expectation from a model's output, based on how sample difficulty works with humans. If a human cannot answer simple questions about a topic, it will not be able to correctly answer more difficult questions on the same topic, so progressively difficult input images to a model should follow a specific pattern in terms of their accuracy.

The authors build a new metric (the hierarchical learning score) together with a dataset of generated synthetic images corresponding to multiple difficulty levels, to assess if image classification models follow difficulty expectations when making predictions. Additionally the paper presents a way to estimate a model's performance by only showing some images instead of the whole dataset, based on how the model performs on a subset of varying difficulty.

Contributions are the new hierarchical learning score metric, the synthetic image difficulty score, results on state of the art image classification models, and a new evaluation methodology based on the GRE to estimate model performance without showing a full dataset, via adaptive evaluation.

**Strengths:**

- The paper is very well written and easy to understand, with excellent presentation of results.
- This paper is very interesting, as it combines expectations from humans in terms of example difficulty and applies it to machine learrning models, providing a novel way to evaluate and see the differences between model's predictions.
- The authors propose an image dataset with difficulty labels that is generated via prompting using Dall-E 3, and it is the base for evaluating image classification models, at least this shows that such dataset is needed to produce the evaluation that this paper proposes.
- Results show that many models do follow the hierarchical learning expectations with respect to example difficulty, which I believe is a novel contribution to the state of the art.
- The paper also proposes adaptive evaluation, where a model is first shown a subset of the data, and depending if the model answers correctly for each difficulty setting, different images are shown in successive rounds, similarly to how GRE and other adaptive tests work, with the idea of reducing the number of samples required for evaluation. I think this idea is novel and interesting.
- For both kinds of evaluation, the paper digs deeper and makes some interesting ablation studies and correlations with model accuracy, which work as validation that the proposed metrics are interesting and usable in the real world.

**Weaknesses:**

- The authors claim that there is no dataset with human difficulty labels, but this is not true. The PASCAL VOC dataset (in particular VOC 2012) has per-bounding box difficulty and occlussion/truncation labels, while this is not the same as easy/medium/hard as this paper proposes and VOC is an older dataset, the VOC dataset does have information to infer a good human difficulty label that could be very useful for this evaluation. Considering that now VOC has some kind of difficulty labels, is it really necessary to generate a synthetic dataset?
- The dataset is generated using Dall-E 3 and the prompts are generated using GPT-4, I have reservations about using a generative model to generate data for evaluation, first from ethical points of view, but more importantly, about validation: first about the GPT-4 prompts representing what the authors truly meant, and second that the generated images truly represent the different difficulty ratings as expected. The paper does not mention if both generated prompts and generated images were validated, only by producing experiments results that are the main results of the paper. Data quality is very important due to being a synthetic dataset and a new metric/benchmark meant to evaluate future models.
- Many results, in particular the ones in Sec 5 about adaptive evaluation, depend on randomness to select a subset of images from the full dataset, and in this case the standard deviation should be reported, specially as there is the potential of cherrypicking in adaptive evaluation, the protocol should be more strict to prevent this in future work that uses adaptive evaluation.
- To me it is not clear why the result in Figure 6, the confidences are averaged across multiple models, the confidences are different for each model, so a distribution of confidences divided per difficulty should be presented for each model individually, not as an aggregate, because this hides the differences between models.

**Questions:**

- How were the generated prompts and generated images validated? It is very possible that some prompts or images are incorrect, the paper does mention this possibility, but its not clear if there were protocols or measures to attempt to prevent incorrect prompts or images from being part of the dataset.
- Why were the confidence scoresin Figure 6 aggregated for all models? It does not make sense to me to do this, as different models produce different confidences, and this should be reflected in the results.
- In adaptive testing, evaluating on less images has the potential of biasing the results, which is not what evaluation should aim for, so what is the real-world use case of adaptive testing?

---

### Official Review · Reviewer_hSYS · 2024-11-03

**Soundness:** 3
**Presentation:** 3
**Contribution:** 2
**Rating:** 6
**Confidence:** 5

**Summary:**

The authors try to identify if computer vision models for image classification behave in a manner similar to humans, who are able to solve hard questions only if they have the ability to solve simpler questions that potentially compose the harder problem. In order to do this, the authors create a dataset using a large language model GPT-4 (to first generate objects, attributes, and difficulty level qualifier words) followed by a text to image generation model DALL-E (to generate the image). The generated dataset is used to evaluate different image classification models pre-trained on ImageNet and LAION, and it is observed that the heirarchical decision making ability is largely followed (non-heirarchical patterns like easy × medium ✓ hard × are not observed much). The authors also extend the evaluation setup to check whether a smaller subset can be utilized for evaluation based on heirarchy and used to approximate the results on the full dataset. This is done by performing evaluations in two rounds: first with 1 easy, 2 medium, 1 hard, and then in the second round based on the results of the first round. The authors show that this adaptive testing framework is able to approximate the full evaluation better than statically choosing three exemplars from each difficulty regime.

**Strengths:**

* The paper is well written and quite self-contained. A simple hypothesis is questioned, a new dataset is presented to evaluate it, and conclusions are consistent with the original hypothesis "Vision Models Develop Human-Like Progressive Difficulty Understanding". The dataset generation and evaluation process is well explained, and each ablation is well motivated.
* The central result of the paper is quite convincing - image classification models seem to follow largely consistent decision rules across difficulty heirarchy as humans. The authors show consistent results on this across three models and two pre-training datasets. While not exactly surprising, the results validate the central hypothesis of the paper quite convincingly.
* The adaptive evaluation method can be a good appoximation for testing synthetic datasets in the future, as the authors demonstrate that the error margins of adaptively choosing samples based on their two-step apporach are better than statically choosing one exemplar from each difficulty. Although there is not much of a sample efficieny gain from this (based on Table 1), the precision gains are significant enough.

**Weaknesses:**

* "Since there does not exist any real data labeled with ground-truth difficulty," this seems like a vast simplification. First of all, the difficulty labels developed by the authors are quite arbitrary and based on their intuitions. A similar heuristic could be developed for existing datasets using summary statistics (e.g. number of objects or number of relations in a scene classification dataset). Secondly, there are datasets that are intentionally curated to serve as difficult counterparts of well known datasets (e.g. ObjectNet, Barbu et al vs ImageNet) and a measure of class-wise difficulty could be derived from drop in performance from class average compared to the reference dataset. Lastly, there are datasets from human cognitive science such as SCEGRAM Öhlschläger et al which have been used in computer vision that explicitly contain images with human cognitive difficulty labels included based on scene type.
* I have a few questions regarding the central question in this paper (do image classification models behave similar to humans when posed increasingly more complex decision problems). Firstly, has this phenomena been reported in image/object classification among humans. If so, what were the results and how do they compare to the results in this paper. Secondly, is there a sense of compositionality in the easy -> hard problems in the human studies (the author's example, 2x3 for easy 2x3x4 for hard has a compositional element). If so, what is the analogue here in the dataset created by the authors? Since the connection to human cognitive science is so important in this paper, I believe these important questions need to be addressed.


## References

1. Barbu, Andrei, et al. "Objectnet: A large-scale bias-controlled dataset for pushing the limits of object recognition models." Advances in neural information processing systems 32 (2019).
2. Öhlschläger, Sabine, and Melissa Le-Hoa Võ. "SCEGRAM: An image database for semantic and syntactic inconsistencies in scenes." Behavior research methods 49 (2017): 1780-1791.

**Questions:**

* Adaptive testing, followed by harder image generation and further testing was first introduced by Gao et al. This paper is quite relevant to the author's work and should be discussed in the "Adaptive model evaluation" section of related works.
* Regarding the confidence based validation of dataset grouped as easy/medium/hard, can the authors expand on what confidence score was used? This is not mentioned in the results (or I might have missed it). I ask this, because some confidence measures such as softmax scores are not very calibrated.


## References

Gao, Irena, et al. "Adaptive testing of computer vision models." Proceedings of the IEEE/CVF International Conference on Computer Vision. 2023.

---

### Official Review · Reviewer_Lx7Q · 2024-11-04

**Soundness:** 1
**Presentation:** 2
**Contribution:** 1
**Rating:** 3
**Confidence:** 4

**Summary:**

This work presents a newly built test dataset for image classification that in addition to the usual ground truth label it also supplies a difficulty level (easy, medium, hard). The dataset is generated by the DALL-E 3 model. The prompts that trigger  DALL-E are generated by GPT-4 that is asked to generate prompts combining 100 classes (a subset of ImageNet), 10 attributes that can make the images difficult (occlusion, quality, size, color, viewpoint, lighting, texture, style, rotation and position) and 3 difficulty levels (easy, medium and hard). The dataset is therefore composed of 36000 images.

Using this dataset the authors explore and verify the hypothesis that models, like humans, often exhibit a hierarchical learning behavior. That is, if a model cannot solve a classification task of a given difficulty typically it cannot solve anything that is even more difficult. Based on this, and inspired by the Graduate Record Examination (aka GRE) for admissions into U.S. universities, the authors develop a dynamic testing mechanism in order to be able to predict a model accuracy without exhaustively using the whole test set. The proposed heuristic is based on two rounds of testing. In the first round the model is shown 1 easy sample, two medium and one hard. Depending on the results of the first round, the difficulty of the 5 images shown in the second round is dynamically changed based on some heuristics. By doing this the model is tested on a total of 9 images only.

**Strengths:**

The dataset introduced could be useful for the community that works on curriculum learning or robustness.
Additionally the paper reads well.

**Weaknesses:**

I think the dataset could be useful but I find the overall work has the following major weaknesses:
- The main hypothesis (models are hierarchical learners) had already been validated (see details below).
- The difficulty label assignment is questionable.
- Adaptive testing is a not well justified heuristic, it could lead to changes in the rank amongst models, and its usefulness is not obvious.
- Some important key details are missing to fully understand the proposed method.

Main hypothesis about model being hierarchical learner already validated.
In a different setting this hypothesis had already been shown. For example Saxena et al. “Data Parameters: A New Family of Parameters for Learning a Differentiable Curriculum” (NeurIPS 19) showed that when the model during training is given the possibility to delay learning of some samples (by giving the model the flexibility of choosing the cross entropy temperature per sample) the easiest images are learned earlier while the more difficult are delayed until later.

Difficulty label assignment.
Asking a model to produce an object with some amount of occlusion (or any other attribute used) likely will correlate to some extent with actual difficulty, however, the label might not be accurate. In fact the results presented in Table 3 and Appendix A4  already show that some attributes are intrinsically more difficult than others but the proposed labeling does not capture these differences. As a result the claim that “This dataset is balanced across types of class, attribute, and difficulty level” (Page 5) is not guaranteed anymore. This could also lead to high variability of the results depending on which attribute is sampled during the dynamic testing. Perhaps a better way of assigning such label could be via a user study. Alternatively, one could use the most accurate model and Monte Carlo drop out: if the image is always predicted correctly it gets the easy label, those images that are always classified incorrectly get the hard label and the other the medium (one could even have a finer grade if they wish). Note that by doing this it is likely that not all attributes will be balanced across difficulty level, hence the authors could continue to generate samples for a given pair category-attribute until they obtained the desired number of images for each difficulty level.

Adaptive testing is a not well justified heuristic, it could lead to changes in the rank amongst models, and its usefulness is not obvious.
Many of the choices in the adaptive testing are heuristic that lack a deeper justification. For example: Why 4 images in the first round? why 5 in the second? Why the choice of the distribution of round 2 is as shown in Figure 5? One could think of many variations but why those numbers specifically? Given the lack of rational for these choices a robustness analysis would have been important.
Additionally, from Table 2 it seems that the error of the proposed method could lead to a rank change among the models.
Lastly, It is possible that I am missing some interesting use case, but I think that fast testing is of limited utility given that usually testing is not the bottleneck. However I could be missing some interesting use cases and be happy to change my opinion on this aspect.

Missing key details.
The paper is well written but I find some important details missing. Here are the main ones:
- Standard deviation for the tests is missing. This is very important in general, but especially due to the point made above about the potential high variability of the results due to inaccurate difficulty label.
- The way confidence was compute in Figure 6 is missing.
- It is unclear how Accuracy is estimated in Table 2 when using adaptive testing. It cannot be the accuracy over the 9 images or it would be unfair for those models that do well in the first round.

Minor Suggestions
- The authors make the following claim in the abstract “Anything else hints at memorization.” Since training sets are publicly available I suggest to verify which is the closes training point to see if there is near duplicate? This would provide evidence to this statement that at the moment can only be a speculative guess.
- Page 2: “learning dynamics” usually refer to what happens during training. Since this method is an analysis after training I would avoid using this term as it can lead to wrong expectation or confusion.

**Questions:**

I believe that the dataset could be useful but the method and the difficulty label might have to be re-thought. These are questions that if answered could improve the paper but I am unsure a simple rebuttal will be sufficient to re-assess the results.

I would like the authors to
- Propose another method for assigning difficulty label (I suggested a couple above) and compare it with the proposed one.
- All experiments should be repeated (arguably more than 3 times) and error bars should be provided.
- Provide justification for the choices of the adaptive testing and provide robustness analysis
- Provide example of use cases for fast testing.
- Provide information about how the confidence was computed.
- Provide information about how Accuracy estimated in Table 2 when using adaptive testing.
- Is it correct to say that according to Table 2 the ranking amongst the models could change when using the full test set compared to the adaptive testing?
- In Figure 6 the authors compute the confidence sliced by difficulty. Why not also compute the actual accuracy sliced by difficulty?
- Why the prompt at Page 5 say “nine level of variability”?

---

### Official Review · Reviewer_wbeS · 2024-11-10

**Soundness:** 1
**Presentation:** 2
**Contribution:** 1
**Rating:** 1
**Confidence:** 5

**Summary:**

This paper investigates to what degree are models similar to humans in terms of their ability to recognize images of varying difficulty. The authors start from the assumption that some problems are are strictly more more difficult for humans then others and apply this to the task of object recognition. To investigate the recognition difficulty similarity between humans and models the authors created a new dataset of dalle-3 generated images using prompts containing different class names, attribute types, and attribute values. The authors evaluated ~12 models on this new dataset and concluded that models exhibit a structured understanding of example difficulty based on the overall accuracy of the set of models across different prompt difficulty levels.

**Strengths:**

- The authors investigate an interesting question
- Their experiments and ideas are presented clearly
- With some human experiments their dataset could potentially be useful for future work

**Weaknesses:**

I believe that the authors set out to investigate an interesting question and I encourage them to keep building on the work in this paper, but the current approach presented has many limitations and is insufficient to answer their questions of interest. First, there is no human evaluation of difficulty of this new dataset. The prompts are chosen with the hope that they lead to more difficult images, but there is on way to know if they are really difficult in the sense that a human would be more likely to classify them incorrectly. I would argue from the example images provided in figure 2 that human accuracy should be close to 100% on this dataset even in the most difficult cases. What this dataset really tests is models robustness to varying different image parameters which has already been explored in previous work using both real and synthetic images ex. ImageNet-C,  ImageNet Rendition, Stylized-ImageNet, ObjectNet. This work is also missing some related work on image difficulty some of which include human evaluations methods the authors may find useful in building on this work: How hard are computer vision datasets? Calibrating dataset difficulty to viewing time (Mayo 2023), Trivial or impossible -- dichotomous data difficulty masks model differences (on ImageNet and beyond) (Meding 2021), Characterizing Structural Regularities of Labeled Data in Overparameterized Models (Jiang 2020), Dataset Difficulty and the Role of Inductive Bias (Kwok 2024)

**Questions:**

Can you justify your choice of prompt parameters and explain how you can know he resulting difficulty level of the images that dalle-3 is creating based on those prompts?

---

### Comment · Area_Chair_kXjr · 2024-11-24

Dear Reviewers,

This is a gentle reminder that the authors have submitted their rebuttal, and the discussion period will conclude on November 26th AoE. To ensure a constructive and meaningful discussion, we kindly ask that you review the rebuttal as soon as possible and verify if your questions and comments have been adequately addressed.

We greatly appreciate your time, effort, and thoughtful contributions to this process.

Best regards,
AC

---

### Comment · Area_Chair_kXjr · 2024-11-27

Dear Reviewers,

We wanted to let you know that the discussion period has been extended to December 2nd. If you haven't had the opportunity yet, we kindly encourage you to read the rebuttal at your earliest convenience and verify whether your questions and comments have been fully addressed.

We sincerely appreciate your time, effort, and thoughtful contributions to this process.

Best,

AC

---

### Meta-Review · Area_Chair_kXjr · 2024-12-13

**Metareview:**

This work proposes a progressive evaluation for image classification based on the premise that a model that fails at a simple task, would fail at a harder version of the task (similar to GRE). The evaluation is generated by prompting DALL-E to generate images with different levels of difficulty like occlusions. With this evaluation, the authors find it is possible to approximate full test performance using just a fraction of the data.

**Strengths**
* The adaptive testing approach, inspired by standardized tests, reduces evaluation steps while maintaining accuracy.
* Reviewers found the paper well-written, self-contained, and effectively illustrated through ablations and visual examples.

**Weaknesses**

* Critical (wbeS, hSYS, Lx7Q, 1Azw): this evaluation is made with a generative model. However, generative models are difficult to control, which makes this evaluation unreliable.
* Medium / Low (Lx7Q, 1Azw): Motivation. This evaluation makes it faster to test a model and it is particularly useful for LLMs, which are slow to evaluate, however, the paper focuses on image classification. Also, I am not sure whether the results and hypotheses in this work transfer to other modalities since one would need to rethink the meaning of difficulty.
* Low: deep learning models are known to learn spurious correlations and to be vulnerable to adversarial attacks. This could make a model fail on a seemingly easy sample while still be able to classify a hard one.

I cannot recommend this work to be accepted at ICLR given that the weaknesses outweigh the strengths. I suggest the authors to work on making their data more reliable and to provide a language version of their framework.

**Additional Comments On Reviewer Discussion:**

After reading the reviews and the author's rebuttal, I see three main concerns remain:

Critical (wbeS, hSYS, Lx7Q, 1Azw): this evaluation is made with a generative model. However, generative models are difficult to control, which makes this evaluation unreliable.
Medium / Low (Lx7Q, 1Azw): Motivation. This evaluation makes it faster to test a model and it is particularly useful for LLMs, which are slow to evaluate, however, the paper focuses on image classification. Also, I am not sure whether the results and hypotheses in this work transfer to other modalities since one would need to rethink the meaning of difficulty.
Low: deep learning models are known to learn spurious correlations and to be vulnerable to adversarial attacks. This could make a model fail on a seemingly easy sample while still be able to classify a hard one.

Reviewer Lx7Q engaged in discussion and agreed with me on the most critical weaknesses.

---

### Decision · Program_Chairs · 2025-01-22

Reject